# Localizing the Center of Surgical Action in Laparoscopic Videos: A Point-Supervised Heatmap Regression Approach

**Amir Ebrahimzadeh**[1] [iD]                    AMIR.EBRAHIMZADEH@MED.UNI-GOETTINGEN.DE
**Nazila Esmaeili**[1]                            NAZILA.ESMAEILI@MED.UNI-GOETTINGEN.DE
**Philipp Horvath**[1]                            PHILIPP.HORVATH@MED.UNI-GOETTINGEN
**Michael Ghadimi**[1]                            MGHADIMI@MED.UNI-GOETTINGEN.DE
**Jannis Hagenah**[1,2]                           JANNIS.HAGENAH@MED.UNI-GOETTINGEN.DE

[1] *Center for Digital Surgery, Department of General, Visceral and Pediatric Surgery, University Medical Center Göttingen, Germany*

[2] *Fraunhofer Research Institution for Individualized and Cell-based Medical Engineering (IMTE), Lübeck, Germany*

## Abstract

Automated assessment of laparoscopic camera centering requires reliable spatial localization of the center of surgical action (COSA). We formalize this task as a point-localization problem addressed via heatmap-regression. We introduce a dataset of 500 annotated laparoscopic video clips and evaluate three encoder-decoder architectures (UNet-ResNet-34, Segformer-MiT-B1, and a custom DINOv3-ViT-S). DINOv3 achieved the best performance, with approximately 95% of detections deviating less than 20% of the frame diagonal from ground truth. This establishes a strong baseline for the COSA localization task and provides an open benchmark for future camera-navigation research.

## 1. Introduction

High-quality laparoscopic camera navigation (LCN) is essential for safe, efficient minimally invasive surgery. Poor navigation, characterized by excessive lateral offset, suboptimal framing, or slow target acquisition, forces the operating surgeon to divert attention toward orienting the view rather than executing the procedure (Huettl et al., 2020; Jarc and Curet, 2017). Structured frameworks for evaluating camera navigation typically decompose quality into several independent criteria; among these, *centering* refers to keeping the center of surgical action (COSA) in the central region of the frame (Huber et al., 2018). Automated centering assessment requires answering a prior geometric question: *where, in image coordinates, is the COSA?* Despite progress in surgical-scene understanding (e.g., phase recognition, instrument detection), COSA localization has received little attention as an independent problem (Khan et al., 2025). Existing segmentation-based approaches are insufficient, as the COSA is not simply the center of mass of all visible instruments, but depends strictly on the relevant active instruments.

**Contributions.** This paper makes the following contributions: (1) We define the *COSA localization* task in a laparoscopic video frame, enabling the automated assessment of surgical centering; (2) We present a dataset of 500 laparoscopic video clips annotated with point-level COSA coordinates at 1 FPS; and (3) We establish a heatmap-regression thoroughly evaluated with three encoder–decoder architectures and introduce an evaluation approach suited to the approximate localization requirements for camera centering assessment.

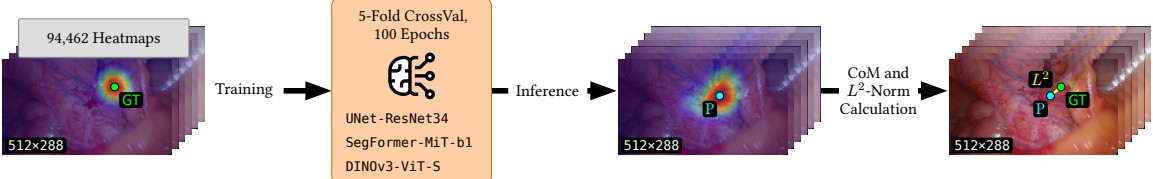

Figure 1: Overview of the proposed COSA localization pipeline (see Figure 3 in appendix).

## 2. Methods

**Dataset and Preprocessing.** The dataset comprises 500 laparoscopic video clips (10 seconds, 30 FPS, 1920 × 1080), sampled without overlap from 50 distinct surgeries performed at University Medical Center Göttingen (UMG). A medical computer-vision expert manually placed a single $(x, y)$ coordinate per frame on 3,547 valid keyframes, denoting the COSA. Annotations were reviewed and approved by a practicing surgeon at the UMG. Assuming only minimal displacement of the surgical action center between consecutive annotations at 1 FPS, we linearly interpolated the annotated $(x, y)$ coordinates across intermediate frames, yielding frame-level annotations at 30 FPS (94 462 frames across 50 clips). The frames are resized (512×288), augmented (horizontal flips ($p = 0.5$), brightness/contrast perturbations ($\pm 0.2$, $p = 0.5$)) using Albumentations (Buslaev et al., 2020), and normalized (ImageNet statistics), with annotations transformed accordingly. Rather than direct coordinate regression, each scaled point annotation is encoded as a 2D Gaussian heatmap on the $512 \times 288$ grid with spread parameter $\sigma = 32$ pixels (see Figure 1). Code and data will be available at https://gitlab.gwdg.de/cds/lcn-cosa-localization upon acceptance.

**Model Architectures and Training.** Three architectures are evaluated to predict the heatmap representing the area of surgical action based on the respective video frame, each accepting a three-channel RGB input and producing a single-channel heatmap in the range $[0, 1]$ ensured by a Sigmoid activation: (1) **UNet-ResNet-34**: standard UNet decoder with an ImageNet-pretrained ResNet-34 encoder (He et al., 2016; Iakubovskii, 2019); (2) **Segformer-MiT-B1**: transformer-based Segformer decoder with a Mix Transformer (MiT-B1) encoder (Xie et al., 2021); (3) **DINOv3-ViT-S**: custom regressor built on a frozen DINOv3-ViT-S/16 backbone (Siméoni et al., 2025). Patch tokens are reshaped into a 2D spatial feature map and upsampled through a lightweight decoder comprising four sequential transposed convolutions with Batch Normalisation and ReLU activations, followed by a $1 \times 1$ convolution with Sigmoid activation. All models are trained with MSE loss and AdamW (encoder: $10^{-5}$; decoder/head: $10^{-4}$), with an adaptive learning rate for up to 100 epochs. We utilize a 5-fold cross-validation scheme, with folds split at the video level to prevent data leakage. COSA coordinates are decoded from predicted heatmaps using a center-of-mass estimator. Performance is quantified using the **Percentage of Correct Keypoints (PCK)**. For each threshold $d \in [0, 587]$ px, we report the fraction of frames whose predicted coordinate lies within a Euclidean distance $d$ of the ground-truth. PCK at $d = 117$ px (approximately 20% of the frame diagonal) is used as the primary scalar summary. A $K$-fold ensemble is evaluated on the independent test set by averaging predicted heatmaps across all five fold models prior to coordinate decoding.

| Model | PCK@0.1 | PCK@0.2 | Mean Dist. (px) |
|---|---|---|---|
| DINOv3-ViT-S | **0.763** | **0.945** | **45.19** |
| SegFormer-MiT-b1 | 0.758 | 0.930 | 46.58 |
| UNet-ResNet34 | 0.699 | 0.909 | 52.04 |

Table 1: PCK and Mean Euclidean Distance Results

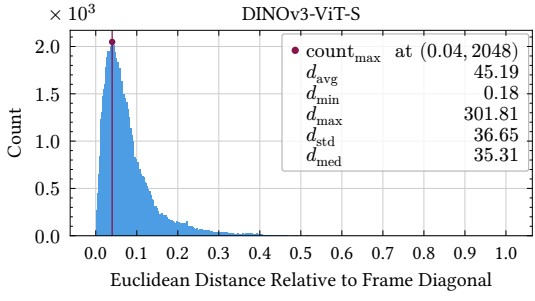 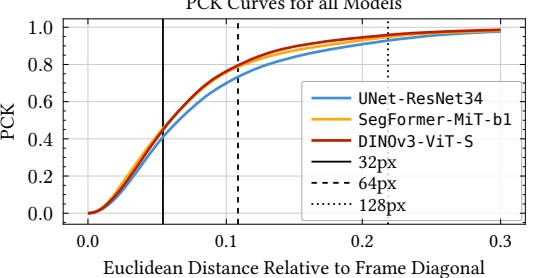

Figure 2: *Left:* Distribution of prediction errors (Euclidean distance in pixels relative to frame diagonal) for all the DINOv3-ViT-S folds including ensemble. *Right:* PCK curves for the three benchmarked architectures.

## 3. Results and Discussion

DINOv3-ViT-S achieved the best performance, yielding a PCK@0.2 of approximately 95% and a mean Euclidean distance error of 45.19 pixels (see Table 1 and Figure 2). Thus, in over 90% of frames, the predicted coordinate lies within 20% of the frame diagonal from the ground truth. We consider this precision sufficient for computing a reliable centering offset. We assume introduced biases through manual selection of sigma and the linear interpolation of 30FPS labels. However, given the error-insensitive metric, we believe that our evaluation accounts for the errors introduced by that. Furthermore, including temporal information might increase the accuracy of the model further.

## 4. Conclusion

To the best of our knowledge, this is the first work addressing automated skill assessment in LCN. We formalized the spatial localization of the center of surgical action as a well-defined computer-vision task, collected a novel annotated dataset, and successfully solved the localization problem. We have demonstrated that automated skill assessment regarding camera centering in LCN is practically feasible. Future work will incorporate temporal context, for example through recurrent or attention-based sequence models, to improve robustness in dynamic surgical scenes. Downstream, the predicted COSA coordinates will be used to derive a continuous centering score by measuring their Euclidean offset from the frame center, closing the loop toward fully automated laparoscopic camera-navigation assessment.

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

# Appendix A. Methods

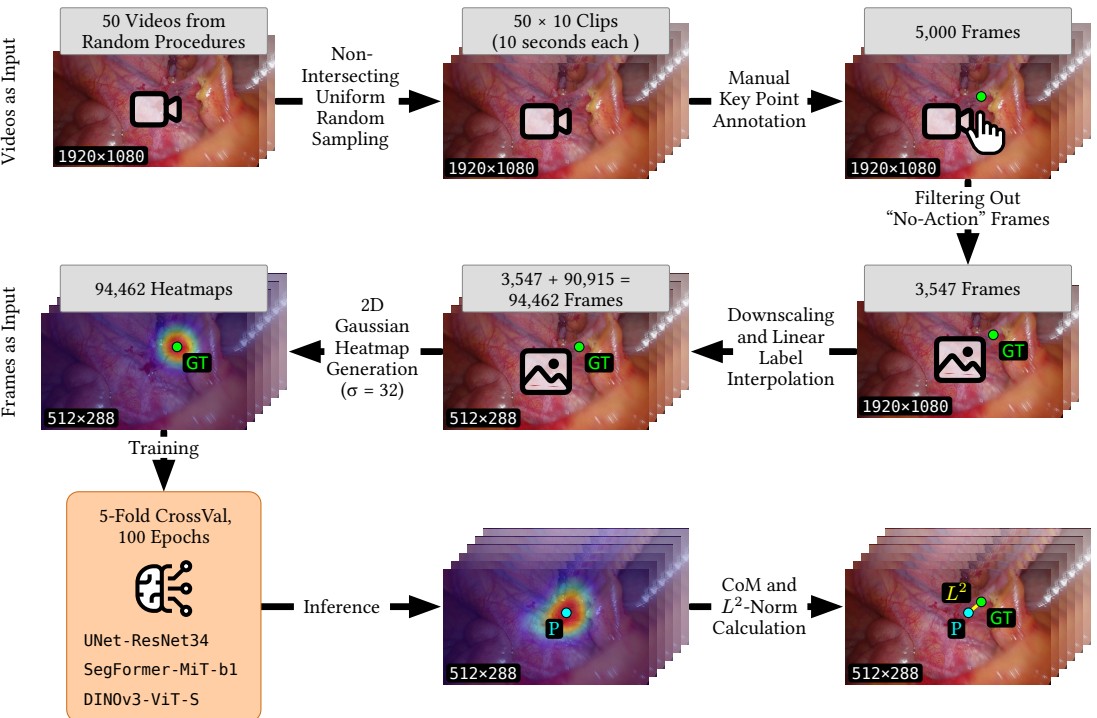

Figure 3: Comprehensive overview of the proposed COSA localization over the entire pipeline (including preprocessing and labeling).

