# OpenReview forum: "Localizing the Center of Surgical Action in Laparoscopic Videos: A Point-Supervised Heatmap Regression Approach"
_MIDL.io/2026/Short_Papers — MIDL 2026 - Short Papers Poster_

### Official Review · Reviewer_4RtF · 2026-05-03
**New application and dataset**

**Rating:** 5
**Confidence:** 4

**Review:**

The paper is well-written and very clear. The problem is motivated clearly, as are the methods and results. The three models chosen are strong baselines, and, together with the data and code sharing, they will likely have an impact. Potential future extensions also include embedding these kinds of methods in direct feedback loops for surgeons during minimally invasive interventions. The main limitation of the paper is the analysis and presentation of results. The results table (Table 1) and figure (Fig. 2) present the same results three times, each in a slightly different way. Fig. 2 (right) shows the cumulative distribution of the histogram in Fig. 2 (left), and Table 1 lists two points in these distributions. Results look good, but only aggregated values across frames in the test set are provided. This leaves questions unanswered, such as:
- Is performance similar across procedures and videos, or are there videos in which the model underperforms, and why?
- Is it easier to detect a COSA close to the center of the frame than towards the borders of the frame? Assuming that surgeons know what they're doing, I'd expect the GT COSA distribution to have a mode around the center of the frame.
- One of the assumptions made during annotation is that there is linear motion between frames sampled at 1 FPS. Is this learned by the model too, or does the model identify other kinds of motion? Does the GT assumption lead to an underestimation of model performance?

One other thing that would have been good to show: The authors say in the Introduction that the COSA is not simply the center of mass of all visible instruments. To verify this, it would have been good to include this approach as a baseline, too.

**Summary:**

The paper describes a new task, identifying the center of surgical action (COSA) in laparoscopic videos. The authors have annotated the COSA in 500 video clips (>90,000 frames) and evaluated three strong landmark localization models for their ability to identify these points. Results look encouraging, data and code will be shared.

**Strengths:**

- This is a very well-written paper, and an interesting read.
- The paper presents a new task, new data, and code that will be shared.
- The authors include strong baseline architectures using convolutional and Transformer-based architectures

**Weaknesses:**

- Repetition in the presentation of results.
- Analysis of results could have addressed some more questions to provide a deeper insight into model performance.
- A baseline including center-of-mass of segmented tools would have been interesting to verify the assumptions made in the Introduction.

**Justification Of Rating:**

Clear paper, interesting story and new task probably leading to impact and others working on this.

---

### Decision · Program_Chairs · 2026-05-08

Accept (Poster)